# Predicting blood lead in Uruguayan children: Individual- vs neighborhood-level ensemble learners

**Seth Frndak**[1]\*, **Elena I. Queirolo**[2], **Nelly Mañay**[3], **Guan Yu**[4], **Zia Ahmed**[5], **Gabriel Barg**[2], **Craig Colder**[6], **Katarzyna Kordas**[1]

**1** Department of Epidemiology and Environmental Health, University at Buffalo, The State University of New York USA, Buffalo, New York, United States of America, **2** Department of Neuroscience and Learning, Catholic University of Uruguay, Montevideo, Uruguay, **3** Faculty of Chemistry, University of the Republic of Uruguay (UDELAR), Montevideo, Uruguay, **4** Department of Biostatistics, University of Pittsburgh USA, Pittsburgh, Pennsylvania, United States of America, **5** Research and Education in eNergy, Environment and Water (RENEW) Institute University at Buffalo, The State University of New York, Buffalo, New York, United States of America, **6** Department of Psychology, University at Buffalo, The State University of New York, Buffalo, New York, United States of America

\* sethfrnd@buffalo.edu

**Data Availability Statement:** The data represented in this manuscript are not publicly available because the Salud Ambiental Montevideo research team does not have IRB approval to publicly share

## Abstract

Predicting childhood blood lead levels (BLLs) has had mixed success, and it is unclear if individual- or neighborhood-level variables are most predictive. An ensemble machine learning (ML) approach to identify the most relevant predictors of BLL ≥2μg/dL in urban children was implemented. A cross-sectional sample of 603 children (~7 years of age) recruited between 2009–2019 from Montevideo, Uruguay participated in the study. 77 individual- and 32 neighborhood-level variables were used to predict BLLs ≥2μg/dL. Three ensemble learners were created: one with individual-level predictors (Ensemble-I), one with neighborhood-level predictors (Ensemble-N), and one with both (Ensemble-All). Each ensemble learner comprised four base classifiers with 50% training, 25% validation, and 25% test datasets. Predictive performance of the three ensemble models was compared using area under the curve (AUC) for the receiver operating characteristic (ROC), precision, sensitivity, and specificity on the test dataset. Ensemble-I (AUC: 0.75, precision: 0.56, sensitivity: 0.79, specificity: 0.65) performed similarly to Ensemble-All (AUC: 0.75, precision: 0.63, sensitivity: 0.79, specificity: 0.69). Ensemble-N (AUC: 0.51, precision: 0.0, sensitivity: 0.0, specificity: 0.50) severely underperformed. Year of enrollment was most important in Ensemble-I and Ensemble-All, followed by household water Pb. Three neighborhood-level variables were among the top 10 important predictors in Ensemble-All (density of bus routes, dwellings with stream/other water source and distance to nearest river). The individual-level only model performed best, although precision was improved when both neighborhood and individual-level variables were included. Future predictive models of lead exposure should consider proximal predictors (i.e., household characteristics).

individual-level data. However, the data to support the findings of this study could be made available upon reasonable request. Researchers can submit a data request, referencing the study, at https://saludambientalmontevideo.org.uy/contacto/.

**Funding:** This work was supported by the National Institute of Environmental Health Sciences (R01ES023423 to AM; R21ES019949 to KK; and R21ES16523 to KK). The funders had no role in study design, data collection and analysis, decision to publish, or preparation of the manuscript.

**Competing interests:** The authors have declared that no competing interests exist.

# 1. Introduction

An estimated 765 million IQ points were lost among children <5 years of age globally due to lead exposure in 2019 [1]. At the same time, global trends show that children's blood lead levels (BLLs) continue to decline. Among U.S. children <6 years of age, mean BLLs decreased from 13.7μg/dL in 1976–1980 [2] to 1.3μg/dL in 2007–2010 [3]. Recent estimates report a mean of 0.80μg/dL among children participating in the 2011–16 National Health and Nutrition Examination Survey (NHANES) [4]. Despite this promising trend, 1 in 3 children (~800 million) worldwide are estimated to have BLLs ≥5μg/dL [5]. Low lead levels are still concerning in children as declines in IQ are found at BLLs <10μg/dL [6, 7]. In response, the Centers for Disease Control lowered the actionable threshold from 5 to 3.5μg/dL [8]. As no safe threshold of lead exposure has been found, prevention of exposure is of public health concern. Identification of high-risk children using predictive modeling could be an important preventive tool.

Predicting geographic areas with a high prevalence of lead-exposed children [9], may be useful for spatially targeted public health interventions [10, 11]. However, individual-level predictive models could also be implemented in clinical settings where testing is limited [12], or when there is suspected continued exposure beyond the age-required testing window (1–2 years in the US). Numerous efforts have been made to predict children's BLLs with questionnaire-based screening tools, but with limited success. A 2019 review of lead screening questionnaires revealed a wide range of success predicting BLLs >10 μg/dL. Authors of the review concluded that screening questionnaires "are not accurate for identifying children with elevated blood lead levels" [13]. An alternative method of predicting BLLs is through machine learning (ML) classifiers. Recently, Potash et al. (2020) demonstrated that random forest (RF) outperformed logistic regression in predicting BLLs ≥6 μg/dL [14]. Liu et al. (2021) used an ensemble ML approach to predict BLL in Australian children <5 years of age [15, 16]. While promising, these predictive ML models do have limitations. Potash et al. 2020, for example, used a high cut-off for lead exposure while Liu et al. 2021 included data from 1991–2015. ML is promising, but predictive models must be updated to predict lower levels of exposure and include more recent data to reflect global declines in child lead levels.

While ML can handle high-dimensional data, more parsimonious models are easier to deploy, and are less likely to be overfit to irrelevant predictors. For example, while neighborhood-level variables are easier to generate and associate with individual child BLLs, they may add noise to models that could benefit from parsimony. Potash et al. (2020) used only neighborhood-level variables [14], while Liu et al. (2021) used both individual- and neighborhood-level predictors. Future predictive models of low-level lead exposure at the individual level might benefit from more focused data inputs. Furthermore, predictive models can be useful in countries where pediatric lead testing is limited. To implement such models, we must test if individual-, neighborhood-level, or a combination of predictors is best for model development.

Leveraging BLLs of ~7 year-old children from Montevideo, Uruguay, this study aimed to: 1) develop a ML model predictive of BLLs ≥2 μg/dL, and 2) identify whether individual, neighborhood or a combination are most relevant for predicting low-level lead exposure. This knowledge will help inform development of datasets and predictive models that can be used for lead screening.

# 2. Materials and methods

## 2.1 Sample

Salud Ambiental Montevideo (SAM) is an ongoing cohort of children residing in the city of Montevideo, Uruguay. Children were recruited as first-graders (average = 6.86 years of age)

between July 23rd, 2009 and June 24th, 2019 from several elementary schools located in areas of suspected toxic metal exposure. Recruitment procedures for children enrolled in 2009–13 can be found elsewhere [17, 18]. Two changes for years 2015–19 should be noted while all other protocols stayed unchanged: 1) the study received approval from the Ministry of Education to recruit children from public elementary schools, 2) study assessments, including blood draws, took place at the SAM research suite at the Catholic University of Uruguay as opposed to children's schools. For this study, 97 children without blood lead measurements, and 17 children without information on household location or residing outside the city limits of Montevideo were excluded. Furthermore, as the limit of detection (LOD) for blood lead measurement improved during data collection, observations with analytical LOD $\geq 2$ µg/dL (n = 139) were removed. This resulted in an analytical sample of 603. When assessing homogeneity between those included and excluded from our analytical sample, there were no differences in age or sex of participants. The additional recruitment of children from public schools after 2015 corresponded to the lowering of analytical limit of detection (LOD) <2 and inclusion of participants from homes of higher socioeconomic status. There were no differences between those included and those excluded by age, gender, or BMI. However systematic differences were observed. Among those excluded, they were from neighborhoods of lower disadvantage, had more household possessions of wealth, and higher maternal education. These differences suggest that those excluded were of generally higher socioeconomic position. However, the scientific interest of including low-level lead exposure thresholds (i.e. <2) for our predictive model outweighs these potential differences. Study protocols were approved by the institutional review boards of the Pennsylvania State University, University at Buffalo, the University of the Republic of Uruguay, and the Catholic University of Uruguay. All caregivers gave their written consent for participation and publication. Children provided assent. Participants were not involved in the design or dissemination of this research.

## 2.2 Blood lead measurements

A trained phlebotomist collected blood samples from children between 8 and 11am, after an overnight fast. Samples were subsequently transported on ice for measurement at the Toxicology Laboratory at the University of the Republic of Uruguay with Atomic Absorption Spectrometry (AAS). During the early enrollment period, 46–68% (years 2009–2012) were analyzed using flame ionization (AAS, VARIAN SpectrAA-55B - limit of detection (LOD) 1.8 µg/dL)(18). Throughout enrollment, graphite furnace AAS was also used (GFAAS/Thermo ICC 3400 –LOD 1.0 µg/dL, 24% 2015–2016 & LOD 0.4 µg/dL, 48%, 2016–2019) (GFAAS/Varian Spectra AA55B –LOD 0.8, 23%, 2010–2013). Laboratory protocols followed standard procedures [19].

## 2.3 Individual-level predictors

A total of 110 predictors were selected *a priori* based on previous literature of lead exposure in children. A complete list of these variables is presented in **Table 1**. Among those, 77 were measured at the individual level, representing household water testing, clinical, nutritional and questionnaire measures. Two household water metals were included: water lead (Pb) and water iron (Fe). Household water was sampled from the kitchen in a 100ml polypropylene container. Water samples were transported to Pennsylvania State University and analyzed using Inductively Coupled Plasma Mass Spectrometry (ICP-MS) with Collision Cell Technology (Thermo Scientific XSERIES2, Bremen, Germany). More detail on water sample collection and analysis is available in a previous publication [20]. Four clinical measures were included: body mass index (BMI), height for age z-score (HAZ), and blood hemoglobin (g/dL). A nurse

**Table 1. Sample characteristics for all variables and number missing (n = 603).**

| Domains and Variables | Descriptive Statistics | Number Missing |
|---|---|---|
| **Individual Level Measures** | | |
| Blood Lead Level ≥2μg/dL (%) | 47.1% | 0 |
| Year of Enrollment | | 0 |
| 2010 | 2.3% | |
| 2011 | 4.3% | |
| 2012 | 10.5% | |
| 2013 | 11.3% | |
| 2015 | 10.6% | |
| 2016 | 17.1% | |
| 2017 | 17.7% | |
| 2018 | 21.7% | |
| 2019 | 4.5% | |
| Child Sex (% Male) | 51.9% | 0 |
| Child Age in Months (Mean ± SD) | 82.3 (6.08) | 1 |
| Hemoglobin (g/dL) (Mean ± SD) | 13.2 (0.9) | 4 |
| Child's BMI kg/m² (Mean ± SD) | 16.8 (2.6) | 7 |
| Child's Height for Age Z-Score (Mean ± SD) | 0.10 (1.1) | 10 |
| Household Water Iron (Fe) Content (ppm) (Median (Q1, Q2)) | 14.1 (9.97, 21.60) | 456 |
| Household Water Lead (Pb) Content (ppm) (Median (Q1, Q2)) | 0.78 (0.38, 1.92) | 468 |
| Number of Siblings (Mean ± SD) | 1.8 (1.6) | 41 |
| Parental Report of Anemia (% Yes) | 8.8% | 43 |
| Parental Report–Child Hospitalized Ever (% Yes) | 45.7% | 38 |
| Parental Report–Child Taken Iron Supplements Last 2 Months (% Yes) | 4.6% | 37 |
| Mother's Education in Years (Mean ± SD) | 8.3 (2.4) | 14 |
| Father's Education | | 72 |
| Any Primary Education | 38.6% | |
| Any Secondary Education | 50.1% | |
| Any Bachelor's or Greater | 11.3% | |
| Mother's Age in Years (Mean ± SD) | 33.5 (7.02) | 22 |
| Father's Age in Years (Mean ± SD) | 36.3 (8.06) | 72 |
| Mother is Employed (% Yes) | 59.8% | 86 |
| Father is Employed (% Yes) | 87.6% | 184 |
| Either Caregiver Smokes (% Yes) | 62.8% | 28 |
| Average Number of Cigarettes the Mother Smokes per Day (Median (Q1, Q2)) | 0 (0, 6) | 45 |
| Average Number of Cigarettes the Father Smokes per Day (Median (Q1, Q2)) | 0 (0, 10) | 127 |
| Times per Month Mother Smoked During Child's Pregnancy | | 53 |
| Never (%) | 78.2% | |
| 1–2 per day (%) | 4.9% | |
| 3–4 per day (%) | 5.8% | |
| 5+ per day (%) | 11.1% | |
| Parent Report Hours per Week Child Plays Outside during Summer | 8.0 (7.2) | 60 |
| Type of Preschool Attended | | 27 |
| Private | 35.9% | |
| Public | 61.1% | |

*(Continued)*

**Table 1.** (Continued)

| Domains and Variables | Descriptive Statistics | Number Missing |
|---|---|---|
| *Did not Attend* | 3.0% | |
| *Number of Household Possessions (Median (Q1, Q2))* | 3 (2, 4) | 31 |
| *Money Spent on Food for the Household Last Month (% <$3,000 UYU)* | 88.1% | 99 |
| *Household is Crowded (>2 Persons per Room) (% Yes)* | 33.9% | 29 |
| *Year House was Built (Mean ± SD)* | 1983 (116.2) | 316 |
| *Family Members Remove Shoes When Entering House (% Yes)* | 6.8% | 31 |
| *Kids <5 Years in the Household (% Yes)* | 41.6% | 0 |
| *Cooking Water is Filtered (% Yes)* | 9.3% | 33 |
| *Use Filtered Drinking Water (% Yes)* | 45.0% | 30 |
| *Family Cultivates or Consumes Food from a Garden? (% Yes)* | 14.6% | 33 |
| *Either Parent Works in a Job with Source of Exposure (% Yes)* | 33.3% | 3 |
| *How Many Years Ago Did You Last Paint the House Inside?* | | 72 |
| *< 6 Months (% Yes)* | 21.9% | |
| *≥ 6 Months—< 1 Year (% Yes)* | 10.6% | |
| *≥ 1 Year—< 3 Years (% Yes)* | 45.8% | |
| *3+ Years (% Yes)* | 15.8% | |
| *Never Painted (% Yes)* | 6.0% | |
| *Windows are Open >5 Hours a Day in Summer (% Yes)* | 63.1% | 29 |
| *Windows are Open >5 Hours a Day in Winter (% Yes)* | 19.3% | 28 |
| *Floors Cleaned > 1 Time per Week (% Yes)* | 91.5% | 28 |
| *Dusting > 1 Time per Week (% Yes)* | 69.5% | 32 |
| *Household Entryways have a Doormat (% Yes)* | 50.5% | 33 |
| *HOME Inventory Score (Mean ± SD)* | 40.3 (10.2) | 94 |
| *Household Monthly Income* | | 73 |
| *< $10,000 UYU* | 22.8% | |
| *$10,000 - $14,999 UYU* | 23.4% | |
| *$15,000 - $24,999 UYU* | 26.0% | |
| *$25,000 + UYU* | 27.7% | |
| *Estimated Total Kilocalories* | 2040.6 (592.9) | 9 |
| *Estimated Dietary Iron (mg/d) (Mean ± SD)* | 9.4 (4.3) | 9 |
| *Estimated Dietary Protein (g/d) (Mean ± SD)* | 67.0 (19.1) | 9 |
| *Estimated Dietary Fat (g/d) (Mean ± SD)* | 65.3 (24.1) | 9 |
| *Estimated Dietary Carbohydrates (g/d) (Mean ± SD)* | 297.2 (101.7) | 9 |
| *Estimated Dietary Fiber (g/d) (Mean ± SD)* | 3.5 (2.3) | 9 |
| *Estimated Dietary Zinc (mg/d) (Mean ± SD)* | 4.8 (2.1) | 9 |
| *Estimated Dietary Calcium (mg/d) (Mean ± SD)* | 656.0 (246.8) | 9 |
| *Estimated Dietary Folate (mg/d) (Mean ± SD)* | 451.2 (197.4) | 9 |
| *Estimated Dietary Vitamin C (mg/d) (Mean ± SD)* | 49.1 (49.3) | 9 |
| *24-Hour Recall of Fruit Consumption (g/d) (Median (Q1, Q2))* | 75 (0, 150) | 9 |
| *24-Hour Recall of Dark Leafy Vegetable Consumption (g/d) (Median (Q1, Q2))* | 0 (0, 0.01) | 9 |
| *24-Hour Recall of Red Orange Vegetable Consumption (g/d) (Median (Q1, Q2))* | 0 (0, 0.01) | 9 |
| *24-Hour Recall of Beans and Peas Consumption (g/d) (Median (Q1, Q2))* | 0 (0, 0.01) | 9 |

*(Continued)*

**Table 1.** (Continued)

| Domains and Variables | Descriptive Statistics | Number Missing |
|---|---|---|
| *24-Hour Recall of Other Vegetables Consumption (g/d) (Median (Q1, Q2))* | 0 (0, 0.01) | 9 |
| *24-Hour Recall of Potato Consumption (g/d) (Median (Q1, Q2))* | 0 (0, 37.5) | 9 |
| *24-Hour Recall of Bread Consumption (g/d) (Median (Q1, Q2))* | 30.0 (5.0, 60.0) | 9 |
| *24-Hour Recall of Grain Consumption (g/d) (Median (Q1, Q2))* | 0 (16.5, 35.0) | 9 |
| *24-Hour Recall of Pasta Consumption (g/d) (Median (Q1, Q2))* | 15 (0, 37.5) | 9 |
| *24-Hour Recall of Red Meat Consumption (g/d) (Median (Q1, Q2))* | 2.5 (0, 60.0) | 9 |
| *24-Hour Recall of White Meat Consumption (g/d) (Median (Q1, Q2))* | 0 (0, 10.0) | 9 |
| *24-Hour Recall of Processed Meat Consumption (g/d) (Median (Q1, Q2))* | 20.0 (0, 75.0) | 9 |
| *24-Hour Recall of Egg Consumption (g/d) (Median (Q1, Q2))* | 0 (0, 11.25) | 9 |
| *24-Hour Recall of Milk Consumption (g/d) (Median (Q1, Q2))* | 250.0 (0, 400.0) | 9 |
| *24-Hour Recall of Cheese Consumption (g/d) (Median (Q1, Q2))* | 0 (0, 3.76) | 9 |
| *24-Hour Recall of Yogurt Consumption (g/d) (Median (Q1, Q2))* | 0 (0, 100.0) | 9 |
| *24-Hour Recall of Soy Product Consumption (g/d) (Median (Q1, Q2))* | 0 (0, 0.01) | 9 |
| *24-Hour Recall of Fat and Oil Consumption (g/d) (Median (Q1, Q2))* | 6.25 (0, 12.0) | 9 |
| *24-Hour Recall of Milk-Based Dessert Consumption (g/d) (Median (Q1, Q2))* | 0 (0, 30.0) | 9 |
| *24-Hour Recall of Sweets Consumption (g/d) (Median (Q1, Q2))* | 31.1 (0, 52.5) | 9 |
| *24-Hour Recall of Pastries Consumption (g/d) (Median (Q1, Q2))* | 51.2 (0, 100.0) | 9 |
| *24-Hour Recall of Fried Potatoes Consumption (g/d) (Median (Q1, Q2))* | 0 (0, 10.0) | 9 |
| *24-Hour Recall of Sweetened Beverages Consumption (g/d) (Median (Q1, Q2))* | 250.0 (0, 450.0) | 9 |
| *24-Hour Recall of Pizza or Dinner Pie Consumption (g/d) (Median (Q1, Q2))* | 90.0 (0, 175.0) | 9 |
| *24-Hour Recall of Sauces Consumption (g/d) (Median (Q1, Q2))* | 0 (0, 0.01) | 9 |
| **Neighborhood Level Measures** | | |
| *Neighborhood Disadvantage (Mean ± SD)* | 1.5 (1.1) | 0 |
| *Square Area (m) in Census Segment Where Child Lives (Median (Q1, Q2))* | 300,482.0 (175,896.0, 491,455.0) | 0 |
| *Number of Persons in Census Segment Where Child Lives (Median (Q1, Q2))* | 2097.0 (1282.0, 2600.0) | 0 |
| *Normalized Difference Vegetation Index (NDVI) 150m Buffer (Median (Q1, Q2))* | 0.32 (0.28, 0.38) | 0 |
| *Normalized Difference Vegetation Index (NDVI) 300m Buffer (Median (Q1, Q2))* | 0.34 (0.30, 0.39) | 0 |
| *Distance to Nearest Informal Settlement (m) (Median (Q1, Q2))* | 117.6 (13.0, 317.9) | 0 |
| *Distance to Nearest Greenspace (m) (Median (Q1, Q2))* | 201.5 (107.6, 334.3) | 0 |
| *Distance to Nearest Feria (m) (Mean ± SD)* | 829.9 (464.9) | 0 |
| *Number of Markets within 1km Buffer (Median (Q1, Q2))* | 1.0 (0, 2.0) | 0 |
| *Number of Markets within 500m Buffer (Median (Q1, Q2))* | 0 (0, 0.01) | 0 |
| *Distance to Nearest Selected Industries (m) (Median (Q1, Q2))* | 859.8 (501.7, 1277.1) | 0 |
| *Density of All Registered Roads of Montevideo (1km Buffer) (Mean ± SD)* | 15.3 (4.3) | 0 |
| *Density of All Registered Roads of Montevideo (500m Buffer) (Mean ± SD)* | 14.0 (4.1) | 0 |
| *Distance to Nearest Bus Route of Montevideo (m) (Median (Q1, Q2))* | 121.5 (30.0, 218.2) | 0 |
| *Density of Bus Routes (1km Buffer) (Median (Q1, Q2))* | 11.40 (6.2, 15.8) | 0 |

*(Continued)*

**Table 1.** (Continued)

| Domains and Variables | Descriptive Statistics | Number Missing |
|---|---|---|
| *Density of Bus Routes (500m Buffer) (Median (Q1, Q2))* | 18.7 (7.6, 32.7) | 0 |
| *Distance to Nearest Gas Station (m) (Mean ± SD)* | 1,110.1 (574.9) | 0 |
| *Density of Gas Stations (1km Buffer) (Median (Q1, Q2))* | 0 (0, 1.0) | 0 |
| *Density of Gas Stations (500m Buffer) (Median (Q1, Q2))* | 0 (0, 0.01) | 0 |
| *Distance to Nearest Waterway (m) (Median (Q1, Q2))* | 274.3 (139.8, 445.4) | 0 |
| *Density of Waterways (1km Buffer) (Median (Q1, Q2))* | 0.87 (0.54, 1.28) | 0 |
| *Distance to Nearest River (m) (Median (Q1, Q2))* | 1412.4 (704.7, 2066.5) | 0 |
| *Distance to Nearest Waste Disposal Container in Montevideo (m) (Median (Q1, Q2))* | 817.4 (525.5, 1333.9) | 0 |
| *Density of Waste Disposal Containers in Montevideo (1km Buffer) (Median (Q1, Q2))* | 1.0 (0, 3.0) | 0 |
| *Density of Waste Disposal Containers in Montevideo (500m Buffer) (Median (Q1, Q2))* | 0 (0, 0.01) | 0 |
| *Percentage of Dwellings in Census Segment with City Network Water Source (Median (Q1, Q2))* | 0.9 (0, 1.0) | 0 |
| *Percentage of Dwellings in Census Segment with a Well Water Source (Median (Q1, Q2))* | 0.001 (0, 0.004) | 0 |
| *Percentage of Dwellings in Census Segment with Stream or Other Water Source (Median (Q1, Q2))* | 0 (0, 0.02) | 0 |
| *Percentage of Dwellings that Dispose Human Waste through City Network (Median (Q1, Q2))* | 0.57 (0, 0.84) | 0 |
| *Percentage of Dwellings that Dispose Human Waste through Septic Tank (Median (Q1, Q2))* | 0.21 (0.06, 0.42) | 0 |
| *Percentage of Dwellings that Dispose Human Waste into Stream or Other (Median (Q1, Q2))* | 0.01 (0, 0.04) | 0 |
| *Dwellings Comprised of Tenants (Mean ± SD)* | 73.01 (46.98) | 0 |

Note. SD–Standard Deviation; Q1 – 25th percentile; Q3 – 75th percentile; UYU–Uruguayan Peso; m–meters; km–kilometers

or nutritionist took three repeated measures of the child's height and weight. The average height and weight were used to calculate the child's BMI and HAZ. Hemoglobin was measured on a drop of blood using a hemoglobinometer (HemoCue, Lake Forest, CA). The 10 dietary measures were collected using two 24-hour dietary recalls administered at least two weeks apart. A trained nutritionist administered the recall to the caregiver, with the child present to aid in recollection; one recall was administered in person and the second by phone. A database of 342 typical Uruguayan foods was used to calculate dietary kilocalories, iron (mg/day), protein (g/day), fat (g/day), fiber (g/day), carbohydrates (g/day), zinc (mg/day), calcium (mg/day), folate (mg/day), and vitamin C (mg/day) for each of the two days, and an average was calculated. A total of 25 food groups (g/day) was created based on individual foods as described previously [21]. Finally, 36 individual-level variables (**Table 1**) were obtained via questionnaire and completed by the caregiver, who answered questions about family socioeconomic status, health history, and household cleaning practices. These included index scores such as number of household possessions and the Home Observation for the Measurement of the Environment Inventory (HOME). The number of possessions was calculated based on ownership of 12 household assets. Exploratory factor analysis (EFA) retained a single factor with 5 possessions, including DVD player, computer, car, washing machine, and landline phone. The HOME inventory was completed by a social worker during a visit to the child's

home. A summary measure of the home environment was calculated (HOME Inventory) [22], with higher scores reflecting greater household enrichment.

## 2.4 Neighborhood-level predictors

The remaining 32 variables were collected at the neighborhood-level, based on household location. These variables were categorized into census segment characteristics, normalized difference vegetation index (NDVI) and distance/density measures. Statistical software R was used to calculate these variables using several packages including *rgeos* [23], *rgdal* [24], *sp* [25], *RStoolbox* [26], and *raster* [27].

Aggregated 2011 census segment characteristics were downloaded from the Intendencia de Montevideo Servicio de Geomática (IMSG) [28]. Participants were assigned aggregated census segment characteristics based on the census segment within which their home was located. Additional information on the census segment database used for our analysis has been previously published [29]. Census segment variables included a neighborhood disadvantage factor; square area of the census segment (meters$^2$); number of persons residing in the census segment, percentage of households in the census segment with 1) city network water source, 2) well water source, and 3) stream or other water source; and percentage of households in the census segment with 1) city network waste disposal, 2) septic tank waste disposal, and 3) stream or other waste disposal.

A multi-date, 150-meter and 300-meter buffer NDVI score was calculated based on 4-band, 3-meter raster files derived from Planet Image [30]. Cloud-free satellite images of Montevideo were downloaded for 6 summer-time dates: December 24, 2018; January 28, 2019; February 19, 2019; December 23, 2019; January 28, 2020; and February 23, 2020. High-resolution data (3-meters) is only available after 2017. NDVI was calculated for each 3-meter raster, on each date, with the standard formula: NDVI = (near infrared band–red band) / (near infrared band + red band). Average area NDVI was then calculated for each date using a 150-meter buffer around the participant's home. Finally, an overall, average NDVI buffer score was calculated for each participant across the 6 dates.

Several distance measures (in meters) were created from publicly available data from Montevideo Municipality website. Locations and boundaries of informal settlements were determined by the Programa de Integración de Asentamientos Irregulares (PIAI). An irregular settlement was defined as an area with 10 or more dwellings built without authorization or permit. Locations and boundaries of public greenspaces (i.e., parks, gardens, landscaped areas) were determined using photogrammetry of aerial photos by the IMSG. A list of industries provided by the IMSG was reduced to only include those with potential for environmental contamination with toxic elements. These 123 industries included: motor vehicle manufacturers, manufacturers of metallic products, and manufacturers of paints and plasticizers. Geolocations of 163 local ferias–open air markets–were gathered by local partners in Montevideo. These ferias are often a source of fresh produce. The IMSG also provides locations of all registered roads, bus routes, waterways, and major rivers in Montevideo. Distance to the nearest bus route was calculated in meters. Waterways and rivers within the city of Montevideo were downloaded from the IMSG. The IMSG identified waterways and rivers using photogrammetry of aerial photos. Rivers were differentiated from waterways by the IMSG based on flow rate.

Densities of ferias, roads, and bus routes around the participant's home were also calculated. The total length for roads, bus routes, and waterways within 500- and 1,000-meter circular buffers around the participant's home was calculated in meters. Density was then calculated as the total length divided by the square area (meters$^2$) of each circular buffer.

Density of rivers was not calculated due to the overall low density of rivers in Montevideo. The density of markets was calculated as the number of ferias within 1-kilometer and 500-meter buffers. The IMSG also maintains locations of gas stations (ANCAP, ESSO, and Petrobras) in the city of Montevideo. The number of gas stations within 1-kilometer and 500-meter buffers were counted as a measure of density. Locations of special containers for disposal of waste including batteries, cans, plastics, and glass were also counted within 500- and 1,000-meter buffers.

## 2.5 Missing data imputation

Missing values were imputed using the *missForest* package in R [31]. *missForest* is favorable compared to other imputation techniques [32] and useful when unable to account for multiple imputed datasets. Random forest for missing data is also robust when predictors have heavy missingness (>75%), as is the case with some of our predictors [33].

The number of trees (*ntree*) was set to 500 and the number of variables at each node (mtry) was set to 40. This was decided after an iterative process of first testing an increasing number of *mtry* (from 2 to 50) and then testing an increasing number of *ntree* (100 to 1,000 stepped by 100). The final Normalized Root-Mean-Square Error (NRMSE) was 0.001% (a measure of imputation error for continuous variables). The Percent Falsely Classified (PFC) was 20.6% (a measure of imputation error for discrete variables).

## 2.6 Generation of three ensemble models

Three ensemble models were generated using different combinations of predictors. Four base classifiers were chosen: generalized linear model (GLM) with an elastic net penalty, random forest (RF), gradient boosting machine (GBM), and deep neural network (DNN). GLM with an elastic net penalty combines the advantages of both the Least Absolute Shrinkage and Selection Operator (LASSO) and Ridge regression to find the optimal penalty factor, shrinking the beta-coefficient of predictors to zero if not associated with the outcome (LASSO) or near zero (Ridge). RF is an aggregation of randomly generated decision trees, useful in classification and non-parametric estimation of regression problems. GBM is an extension of RF which aggregates decision trees with fewer branches, using classification performance to inform subsequent decision trees. DNN uses neural networks to optimize prediction forward and back-propagation to weight nodes based on classification performance. A stacked ensemble is a method of combining the predictions of all base classifiers using a meta-GLM, improving classification performance and preventing over-fitting. Further discussion of these algorithms and stacking methods can be found in James et. al. 2013 [34].

An individual-level only ensemble was estimated using only the 77 individual-level predictors (Ensemble-I). A neighborhood-level only ensemble was estimated using only the 32 neighborhood-level predictors (Ensemble-N). Lastly, all 109 predictors were used to estimate Ensemble-All. The following process was repeated for each of the three ensembles using the same random seed to ensure data splitting was replicated across combinations of predictors.

First, three randomly split training (50%), validation (25%), and test (25%) datasets were created. Four base classifiers were used to create the ensemble: a GLM, RF [35], GBM [36], and DNN [37, 38]. Optimization and final selection of the hyper-parameters for each base learner was based on the greatest area under the curve (AUC) for the receiver operating characteristic (ROC). Model calibration for all base learners was performed using a grid search for optimal hyper-parameters with 10-fold cross validation on the training dataset. Following the example laid out by Ahmed et al. 2021 [39], 0.001 was set as the stopping tolerance and 2 as the stopping rounds. Stopping tolerances help prevent overfitting, improving overall classification.

Hyperparameters for each base model in the results are also reported as in Ahmed et al. 2021. The best-performing base learner based on highest AUC in the validation dataset was subsequently incorporated into a stacked ensemble. Stacking creates a meta-dataset containing the predictions of each base learner and uses a subsequent meta-classifier to create an ensemble prediction [40]. A generalized linear model (GLM) was chosen as the meta-classifier for the stacked ensemble. Tuning of the stacked ensemble was performed using 10-fold cross-validation. Model creation was implemented using the *h2o* package in R [41].

### 2.7 Model performance metrics

Performance of the final three ensembles (Ensemble-I, Ensemble-N, and Ensemble-All) was assessed using area under the curve (AUC), precision, sensitivity, and specificity in the test dataset. AUC ranges from 0 to 1 and is based on the ROC curve. An AUC of 1 indicates perfect classification. The other performance metrics were calculated from the confusion matrix of predicted classifications based on the F1-optimal threshold. The confusion matrix is a 2x2 table of classifications including true positives (TP), false positives (FP), false negatives (FN) and true negatives (TN). Formula for the F1 score [41] is:

$$F1 = 2\left(\frac{(precision)(sensitivity)}{precision + sensitivity}\right)$$

Where precision is defined as TP/(TP+FP) and sensitivity is TP/(TP+FN). F1 scores range from 0 to 1, with 1 indicating all positives and negatives were correctly classified. Specificity is calculated as TN / (FP+TN), and when high is meant to optimize true detection of a non-event.

### 2.8 Variable importance

Explainable ML was implemented using the *DALEX* package in R to derive variable importance with the test dataset [42]. First, permutation-based calculations were used to estimate variable importance, which reflects the classification performance of the learner if that variable was removed. This method randomly reorders the variable of interest, recalculates the prediction, and the drop in learner performance is recorded (dropout loss). Permutation-based variable importance can be useful when comparing variable importance across many learners as in an ensemble (15). Variable importance statistics were calculated using 1—Area Under the Curve (AUC) as the dropout loss function.

## 3. Results

### 3.1 Sample characteristics

A complete list of variables with sample characteristics is presented in **Table 1**. Children were 6.9 years of age on average, and almost evenly distributed by sex (52% male). Recruitment varied slightly over time from 2010 to 2019 (range 5%-18%), with the most children recruited in 2018. Children had an average BMI of 16.7 (SD 2.6 kg/m$^2$) and hemoglobin level indicative of absence of anemia (Mean ± SD 13.2 ± 0.9 g/dL). Many children came from homes of low socioeconomic status with a low number of possessions of wealth (Median = 3, (Interquartile Range IQR 2–4)), and ~23% of households with monthly income < $10,000 Uruguayan Pesos (~$370.00 USD in June 2015) [43].

Children in SAM came from more disadvantaged neighborhoods (1.5 ± 1.1), compared to the overall city mean neighborhood disadvantage score of 0. Median distance to a bus route was 121.5 meters (IQR 107.6–334.3). Median distance to a waterway was 274.3 meters (IQR

139.8–445.4). Median distance to an industry using toxic metals in production was 859.8 meters (IQR 501.7–1277.1). Children also lived within a median of 201.5 meters to the nearest greenspace (IQR 107.6–334.3) and the median 150-meter buffer NDVI was (Median 0.32, IQR 0.28–0.38). Close to half of the children had BLLs $\geq$2 µg/dL (47%).

## 3.2 Ensemble performance

Model performance was reported during training, validation and testing for all base classifiers and the stacked ensemble in **Table 2**.

### 3.2.1 Ensemble-I

Hyperparameters for the best base classifiers in Ensemble-I: (GLM (AUC test = 0.61): alpha of 0.0, lambda of 2.54, RF (AUC test = 0.72): 50 trees, column sample rate of 0.59, max depth of 20; GBM (AUC test = 0.75): 100 trees, column sample rate of 0.40, max depth of 2, 1 minimum number of rows; DNN (AUC test = 0.68): 3 hidden layers with 100 units, a "Maxout" activation function). ROC curves for Ensemble-I and all base classifiers on the test dataset are presented in **Fig 1A**. AUC in the test dataset for Ensemble-I was 0.75. At the F1-optimal threshold, precision was 0.56, sensitivity 0.79, specificity 0.65.

Variable importance for Ensemble-I is presented in **Fig 2A**. Top ten predictors included: year of enrollment, household water Pb (ppm), HOME inventory Score, household water Fe (ppm), height for age z-score, fruit consumption (g/d), type of preschool attended, household crowding, child age in months, money spent on food last month.

**Table 2. Model Performance across all base classifiers and stacked ensemble in the training, validation, and test datasets.**

| | Ensemble-I | | | | Ensemble-N | | | | Ensemble-All | | | |
|---|---|---|---|---|---|---|---|---|---|---|---|---|
| | AUC | Prec. | Sens. | Spec. | AUC | Prec. | Sens. | Spec. | AUC | Prec. | Sens. | Spec. |
| GLM | | | | | | | | | | | | |
| Training | 0.77 | 0.19 | 0.94 | 0.55 | 0.60 | 0.0 | 0.0 | 0.50 | 0.79 | 0.15 | 0.86 | 0.53 |
| Validation | 0.63 | 0.29 | 0.69 | 0.60 | 0.56 | 0.05 | 0.75 | 0.56 | 0.61 | 0.26 | 0.67 | 0.59 |
| Test | 0.61 | 0.19 | 0.94 | 0.55 | 0.49 | 0.0 | 0.50 | 0.0 | 0.57 | 0.15 | 0.86 | 0.53 |
| RF | | | | | | | | | | | | |
| Training | 1.0 | 1.0 | 1.0 | 1.0 | 0.98 | 0.87 | 0.95 | 0.90 | 1.0 | 1.0 | 1.0 | 1.0 |
| Validation | 0.73 | 0.31 | 0.76 | 0.62 | 0.50 | 0.05 | 1.0 | 0.56 | 0.74 | 0.60 | 0.74 | 0.71 |
| Test | 0.72 | 0.44 | 0.83 | 0.62 | 0.53 | 0.11 | 0.69 | 0.51 | 0.73 | 0.52 | 0.87 | 0.65 |
| GBM | | | | | | | | | | | | |
| Training | 0.86 | 0.66 | 0.86 | 0.76 | 0.77 | 0.55 | 0.68 | 0.74 | 0.86 | 0.65 | 0.89 | 0.76 |
| Validation | 0.77 | 0.56 | 0.80 | 0.71 | 0.50 | 0.10 | 0.75 | 0.57 | 0.74 | 0.63 | 0.74 | 0.73 |
| Test | 0.75 | 0.59 | 0.82 | 0.68 | 0.59 | 0.01 | 1.0 | 0.50 | 0.70 | 0.52 | 0.84 | 0.65 |
| DNN | | | | | | | | | | | | |
| Training | 1.0 | 0.99 | 1.0 | 0.99 | 0.77 | 0.43 | 0.78 | 0.65 | 1.0 | 0.97 | 1.0 | 0.97 |
| Validation | 0.67 | 0.06 | 0.80 | 0.56 | 0.52 | 0.05 | 0.75 | 0.56 | 0.60 | 0.24 | 0.68 | 0.59 |
| Test | 0.68 | 0.32 | 0.86 | 0.58 | 0.49 | 0.03 | 1.0 | 0.50 | 0.61 | 0.24 | 0.73 | 0.54 |
| Ensemble | | | | | | | | | | | | |
| Training | 0.98 | 0.97 | 0.91 | 0.97 | 0.79 | 0.73 | 0.72 | 0.76 | 1.0 | 1.0 | 0.99 | 1.0 |
| Validation | 0.76 | 0.53 | 0.79 | 0.69 | 0.50 | 0.05 | 0.75 | 0.56 | 0.74 | 0.52 | 0.80 | 0.69 |
| Test | 0.75 | 0.56 | 0.79 | 0.65 | 0.51 | 0.0 | 0.0 | 0.50 | 0.75 | 0.63 | 0.79 | 0.69 |

Note. GLM–Generalized Linear Model; RF–Random forest; GBM–Gradient Boosting Machine; DNN–Deep Neural Network; AUC–Area under the curve

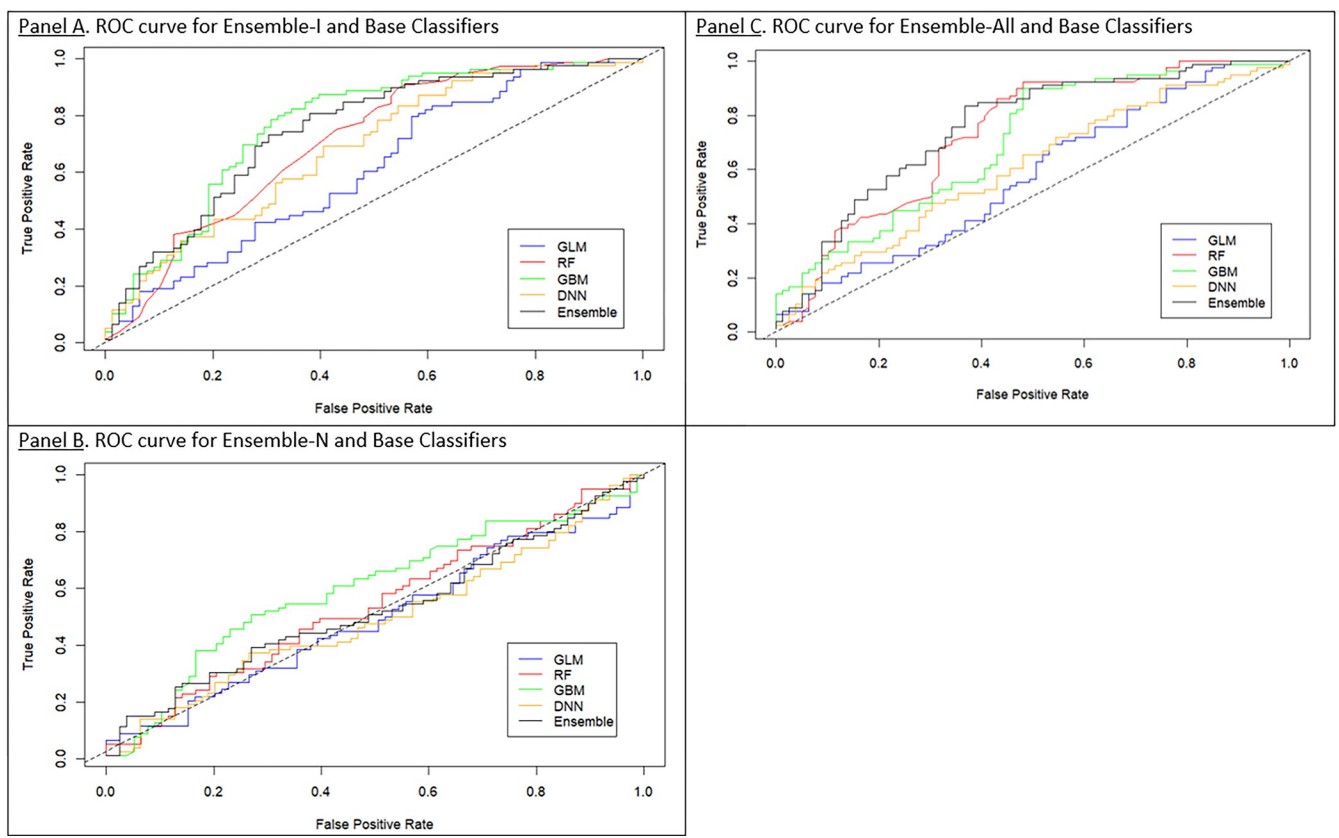

**Fig 1. Receiver operating characteristic (ROC curves for Ensemble-I, Ensemble-N, and Ensemble-All across base classifiers on the test dataset.**

**3.2.2 Ensemble-N.** Hyperparameters for the base classifiers in Ensemble-N: (GLM (AUC test = 0.49): alpha of 0.69, lambda of 0.1, RF (AUC test = 0.53): 4,150 trees, column sample rate of 0.19, max depth of 40; GBM (AUC test = 0.59): 1,000 trees, column sample rate of 0.80, max depth of 18, 100 minimum number of rows; DNN (AUC test = 0.49): 4 hidden layers with 10 units, a "Tanh" activation function). ROC curves for Ensemble-N and all base classifiers are presented in **Fig 1B**. AUC for Ensemble-I was 0.51. At the F1-optimal threshold, precision was 0.0, sensitivity 0.0, specificity 0.50.

Variable importance for Ensemble-N is presented in **Fig 2B**. Top ten predictors included: percent dwellings that dispose waste into the city network, percent dwellings with city network water source, distance to nearest feria (m), density of bus routes (1km buffer), distance to nearest irregular settlement (m), density of all registered roads (1km buffer), distance to nearest gas station (m), distance to nearest greenspace (m), density of waterways (1km buffer), and percent of dwellings with a stream or other water source.

**3.2.3 Ensemble-All.** Hyperparameters for the base classifiers in Ensemble-All: (GLM (AUC test = 0.57): alpha of 0.0, lambda of 4.80, RF (AUC test = 0.73): 310 trees, column sample rate of 0.68, max depth of 50; GBM (AUC test = 0.70): 1,000 trees, column sample rate of 0.80, max depth of 18, 100 minimum number of rows; DNN (AUC test = 0.61): 3 hidden layers with 50 units, a "Maxout" activation function). ROC curves for Ensemble-All and all base classifiers are presented in **Fig 1C**. AUC for Ensemble-All was 0.75. At the F1-optimal threshold, precision was 0.63, sensitivity 0.79, specificity 0.69.

Variable importance for Ensemble-All is presented in **Fig 2C**. Top ten predictors included: year of enrollment, household water Pb (ppm), density of bus routes (1km buffer), HOME

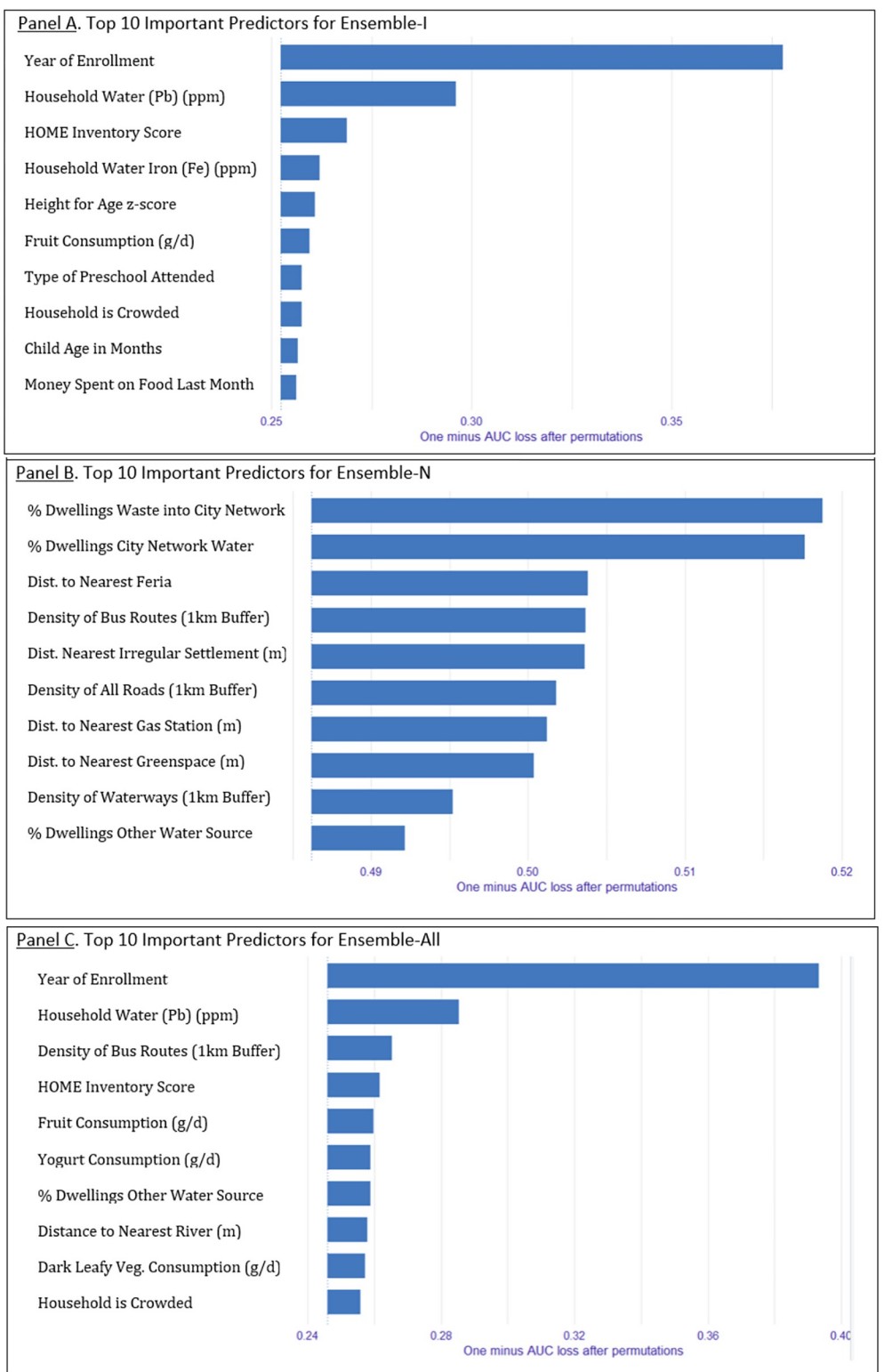

**Fig 2. Variable importance for Ensemble-I, Ensemble-N, and Ensemble-All (1-AUC) in the test dataset.**

Inventory score, fruit consumption, yogurt consumption, percent dwellings with stream or other water source, distance to the nearest river (m), dark leafy vegetable consumption (g/d), and household crowding.

## 4 Discussion

Even at relatively low levels, lead can impact children's cognition and behavior. Thus, predicting which children may have BLLs ≥2μg/dL using recently collected data is of public health importance. By predicting which children are likely to have lead levels above the CDC threshold, public health experts may even be able to prevent exposure, rather than relying on blood tests after exposure has already occurred. However, it is unclear what combination of variables (individual-level, neighborhood-level, or both) are best for BLL prediction. Three different combinations of predictors were used when generating the ensemble classifiers: individual-level only predictors (Ensemble-I), neighborhood-level only predictors (Ensemble-N) and both types of predictors (Ensemble-All). Overall, Ensemble-I and Ensemble-All had similar AUC in the test dataset within two decimal places (0.75), followed by Ensemble-N (0.51). The severe underperformance of Ensemble-N suggests that individual-level predictors may perform better, or Ensemble-N did not successfully capture an important neighborhood-level predictor. While Ensemble-All and Ensemble-I performed similarly, Ensemble-All did have better precision (0.63 vs 0.56, respectively). While studies have shown promising predictive power based exclusively on neighborhood-level predictors, this study suggests that using either individual-only or both individual- and neighborhood-level variables performs best for predicting BLLs ≥2μg/dL.

Collecting more proximal predictors (i.e., household water lead and diet) or unique contextual exposures such as pottery glaze [44] may enhance performance of predictive models of low-level lead exposure. Wilson et al.'s US-based 2023 study used housing conditions as a proximal predictor of childhood lead exposure including conditions of roofs, windows, doors, and exterior paint. The Least Absolute Shrinkage and Selection Operator (LASSO) model selected variables such as condition of the home's porch, exterior paint, windows and doors, roof, and foundations and walls as successful predictors of child BLL ≥3.5μg/dL.[45] A Bayesian network machine learning model published in 2022 achieved similar, though slightly higher accuracy (74%) for predicting child lead exposure ≥2μg/dL. This model used geographic predictors derived from neighborhood water characteristics[46]. Furthermore, unique exposure sources may be considered in predictive models. In 2023, Hoover et al. found that each 14% increase in household firearm ownership rate was associated with 41% higher prevalence of BLL ≥5μg/dL [47].

In Ensemble-I, socioeconomic status, diet, and physical dwelling characteristics were most important for prediction. In general, lower socioeconomic status is directly associated with BLLs in children [48, 49]. Diet is also associated with BLLs in children and may be especially relevant for low-level exposure [50]. Specifically, fruit consumption was important for prediction. Citrus fruit are rich in vitamin C, which has been theorized to be associated with lower blood lead levels [51], although clinical trials have been inconclusive [52]. Finally, household water lead, HOME inventory score, and crowding, all related to housing quality, may be associated with BLLs in children [53]. Curation of data that includes proximal variables such as diet, household characteristics and socioeconomics may enhance low-level BLL prediction.

In Ensemble-I and Ensemble-All year of enrollment was the most important predictor. BLLs are decreasing in children worldwide and it is not uncommon to note secular trends in children's BLLs. The SAM cohort does reflect secular trends of globally decreasing BLLs, although the BLLs in the SAM cohort do still reflect neurotoxic effects [54]. Liu and colleges

(Australia) reported year of enrollment as the most important predictor of BLLs in Australian children [15]. In a recent publication that used ML Bayesian networks to relate water lead and BLL in North Carolina, US, year of blood draw was also the most important predictor of child BLLs $\geq 2$ μg/dL [46]. The importance of enrollment year also suggests that the relevance of predictive models depends strongly on the recency of the data collected. Because predictive models require a large sample size, most studies will include data across multiple years. An indicator variable for year should be part of the predictors being tested, and future studies that have sufficiently large samples, may proceed by testing predictors in each study year separately to understand if other key predictors change over time. Our sample size did not allow us to perform such tests.

It is also difficult to infer from this data what may be changing in the environment across years. First, a cross-sectional study is unable to evaluate changes in BLLs over time. Second, neighborhood-level variables were collected at a single time point, thus it is not possible to evaluate how changes in the neighborhood environment may be related to BLL over time. To further probe these results, year of enrollment was added to the Ensemble-N variable set. While predictive power was improved, Ensemble-N still did not perform as well as Ensemble-I or Ensemble-N.

Some caution is warranted in interpreting these findings. First, the most important predictor in Ensemble-I was household water Pb. In SAM, household water Pb was only minimally associated with urinary and blood lead levels in SAM, dependent on children's iron status (20). In Ravenscroft et al. 2018, however, the analyses were causal in nature, adjusting for socioeconomic status indicators, thereby obtaining the effect estimates adjusted for confounding. In ML, variable importance cannot be interpreted in the same manner as effect estimates in causal models. Complex interactions between predictors can affect overall importance [55, 56]. It should also be noted that precision (or positive predictive value) was low across models (Ensemble-I precision 0.56), while sensitivity (or negative predictive value) was high (Ensemble-I sensitivity 0.79). Thus, the model's ability to rule out children without BLLs $\geq 2$μg/dL was strong, but positive prediction of BLLs $\geq 2$μg/dL was poor. This further suggests that the previous work was correct in asserting that household water Pb, and red meat consumption were not strongly associated with BLL, and the predictive model is likely using these predictors to correctly classify children with BLLs $<2$μg/dL.

There are additional limitations in this study. First, only four popular, high-performing base classifiers were considered. The chosen classifiers may not fit the data well and therefore, some important predictors may not have been selected. Also, not all potentially relevant predictors were included. For example, soil lead levels are known to contribute to children's BLLs [57]. It is important to note that predictive models utilizing area-level characteristics will necessarily be context specific and may not translate well to other samples. Geographic and neighborhood predictors may be more relevant in the U.S. where geographic segregation has contributed to BLLs in U.S. children [58]. Lastly, some of the variables, including household water iron and water lead had a high number of missingness. The predictive model may have been improved with less missing data.

As for strengths, this is the first study to attempt ensemble ML-based prediction of child BLLs $\geq 2$μg/dL with a large set of individual- and neighborhood-level variables. In total, the model included 109 variables (77 individual- and 32 neighborhood-level). Furthermore, uncommonly used variables were also included such as food and nutrient consumption, and home cleaning practices. Furthermore, such sophisticated predictive models are underemployed for predicting toxicant exposure in the Global South. Lastly, the AUC for Ensemble-I was acceptable, though predictive performance could be enhanced.

## 5. Conclusions

In summary, an ensemble ML with only neighborhood-level variables severely underperformed. Ensembles with individual-level variables or a combination of both are ideal. Future predictive models in global contexts may learn from these findings by including more proximal predictors of socioeconomics, diet, and household characteristics. Furthermore, future studies with sufficiently robust sample sizes could leverage multiple years of recently collected data to determine if key predictors change over time, or if predictive models are equally predictive across years.

## Supporting information

**S1 Checklist.**
(DOCX)

## Author Contributions

**Conceptualization:** Seth Frndak, Zia Ahmed, Katarzyna Kordas.

**Data curation:** Seth Frndak, Elena I. Queirolo, Nelly Mañay, Gabriel Barg, Katarzyna Kordas.

**Formal analysis:** Seth Frndak.

**Funding acquisition:** Nelly Mañay, Katarzyna Kordas.

**Investigation:** Elena I. Queirolo, Nelly Mañay, Gabriel Barg, Katarzyna Kordas.

**Methodology:** Seth Frndak, Guan Yu, Katarzyna Kordas.

**Project administration:** Elena I. Queirolo, Nelly Mañay, Gabriel Barg, Katarzyna Kordas.

**Resources:** Elena I. Queirolo, Katarzyna Kordas.

**Supervision:** Elena I. Queirolo, Gabriel Barg, Craig Colder, Katarzyna Kordas.

**Writing – original draft:** Seth Frndak.

**Writing – review & editing:** Elena I. Queirolo, Nelly Mañay, Guan Yu, Zia Ahmed, Gabriel Barg, Craig Colder, Katarzyna Kordas.

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
