## [Decision Letter · Decision Letter 0]

26 Mar 2024

PGPH-D-23-01421

Predicting Blood Lead in Uruguayan Children: Individual- vs Neighborhood-Level Ensemble Learners

Dear Dr. Frndak,

Thank you for submitting your manuscript to PLOS Global Public Health. After careful consideration, we feel that it has merit but does not fully meet PLOS Global Public Health’s publication criteria as it currently stands. Therefore, we invite you to submit a revised version of the manuscript that addresses the points raised during the review process.

Please provide evidence from the recent literature, and the conclusion section needs to be expanded for greater impact. Please examine the public health implications of the lead exposure findings and include them in the discussion. Methods should contain the details need to be clearer: For participants excluded due to missing data, it should be evaluated whether the available data for these participants are homogeneous with those of included participants. Different analyzers were used for BLL measurements, their results should be evaluated for transferability to ensure consistency. Did the models should considered parents' occupation as a potential variable, given that parents could bring lead-contaminated clothing from the workplace, affecting children's BLL.

Also, please ensure that tables and figures are explained independently. Explain the acronyms while using for first time

We look forward to receiving your revised manuscript.

Kind regards,

Giridhara R Babu, MBBS, MPH, PhD

Academic Editor

Journal Requirements:

Additional Editor Comments (if provided):

Reviewers' comments:

Reviewer's Responses to Questions

**Comments to the Author**

1. Does this manuscript meet PLOS Global Public Health’s publication criteria? Is the manuscript technically sound, and do the data support the conclusions? The manuscript must describe methodologically and ethically rigorous research with conclusions that are appropriately drawn based on the data presented.

Reviewer #1: Yes

Reviewer #2: Yes

2. Has the statistical analysis been performed appropriately and rigorously?

Reviewer #1: Yes

Reviewer #2: Yes

3. Have the authors made all data underlying the findings in their manuscript fully available (please refer to the Data Availability Statement at the start of the manuscript PDF file)?

Reviewer #1: Yes

Reviewer #2: Yes

4. Is the manuscript presented in an intelligible fashion and written in standard English?

Reviewer #1: Yes

Reviewer #2: Yes

5. Review Comments to the Author

Reviewer #1: The manuscript is well written and the study design is sound. Please find minor comments embedded in the manuscript test. The minor comments that I made are as follows:

1. All acronyms/abbreviations should be defined in full at first use and used consistently thereafter. Please review throughout the manuscript.

2. 97 participants were excluded because of missing data. Were the available data for these excluded participants evaluated for homogeneity with those of included participants?

3. Different analysers were used for the analysis of BLL. Were results from these different analyzers evaluated for transferability?

4. The manuscript reports mean age as ~7years. One would have assumed that exact ages were recorded and the actual mean/median ages should be reported instead

5. Some of the data reported as mean±SD but the SD's are very large implying that the data might have been non normal and therefore needed to be reported as median(IQR). Could the authors please verify the distribution of their data and if non-parameteric use the appropriate summary statistics

6. The year of enrolment was reported to be a strong predictor of BLL. Weren't the years of enrollment also associated with with different BLL measurement techniques? If that was the case, what impact would this have had on your findings?

7. Did your models also take into account the parent's occupation based on the possibility that parents could bring contaminated clothing from the work place etc?

8. Tables and figures should be self contained with all acronyms/abbreviations explained in full

Reviewer #2: I command the authors for doing this valuable research. There are a few changes/improvements that I would like to see before publishing this manuscript as follows:

1. The authors should refrain from using active voices like I and we in the manuscript.

2. Although the main aim of the work is related to prediction, I strongly recommend the authors elaborate more on the lead found in the sample of children involved. PLOS Global Public Health journal, as can be understood from the title, is concerned more in public health than in pure ML applications. If the authors published another paper on these results, summary of the results can be given in the current paper and that published paper should be referred to in this manuscript.

3. I suggest the authors add a couple of more recent references (after 2020) especially (but not necessarily) from PLOS journals.

4. "and evenly distributed by sex" Line 271 on page 7 should be "and almost evenly distributed by sex" or something similar to this statement. 52-48% is not exactly evenly distributed.

5. The conclusions section is way too brief. The authors should elaborate more on the results in the conclusions section.

6. PLOS authors have the option to publish the peer review history of their article (what does this mean?). If published, this will include your full peer review and any attached files.

**Do you want your identity to be public for this peer review?** For information about this choice, including consent withdrawal, please see our Privacy Policy.

Reviewer #1: No

Reviewer #2: **Yes: **Salaheddine Bendak

---

## [Decision Letter · Decision Letter 1]

24 Jun 2024

PGPH-D-23-01421R1

Predicting Blood Lead in Uruguayan Children: Individual- vs Neighborhood-Level Ensemble Learners

Dear Dr. Frndak,

Thank you for submitting your manuscript to PLOS Global Public Health. After careful consideration, we feel that it has merit but does not fully meet PLOS Global Public Health’s publication criteria as it currently stands. Therefore, we invite you to submit a revised version of the manuscript that addresses the points raised during the review process.

We look forward to receiving your revised manuscript.

Kind regards,

Meghnath Dhimal, Ph.D.

Academic Editor

Journal Requirements:

2. We have noticed that you have uploaded Supporting Information files, but you have not included a list of legends. Please add a full list of legends for your Supporting Information files after the references list.

Additional Editor Comments (if provided):

Please address comments of reviewer below. Please provide justification if all comments can not be addressed at this stage

1. On authors’ affiliations Lines 10-11, Department is spelts as ‘Departament’ in two instances.

2. Lines 272-282: I made a comment that the standard way to present means should mean±SD and I also highlighted that very large standard deviations tend to be associated with non-parametric data and in such cases, the mean and interquartile will be recommended as measures of central tendency. This was not addressed.

a. Children had low to average BMI (M = 16.7, SD 273 = 2.6 kg/m2) (What does low to average mean in this case?)

b. Children in our sample came from more disadvantaged neighborhoods (mean = 1.5, SD = 1.1) (What does the mean and SD refer to in this sentence?

c. (mean = 143.6, SD = 129.0), within 400 meters of a waterway 280 (M = 360.0, SD =446.1), and within 1 kilometer of an industry using in toxic metals in production (M = 937.4, 281 SD = 608.1). Children also lived, on average, within 300 meters of greenspace (M = 246.3, SD = 208.4) (These Standard deviations are very high implying non-normal data. It is normally recommended to present such data as median(IQR)

3. In Table 1:

a. Two variables do not have any data entered HAZ and WAZ. Please consider removing these if the data are not there.

b. Hemoglobin is not measured in serum but whole blood. I suggest the serum prefixed to hemoglobin be removed.

4. Except for ‘Departament’ all these comments were made during the previous review cycle

Reviewers' comments:

Reviewer's Responses to Questions

**Comments to the Author**

1. If the authors have adequately addressed your comments raised in a previous round of review and you feel that this manuscript is now acceptable for publication, you may indicate that here to bypass the “Comments to the Author” section, enter your conflict of interest statement in the “Confidential to Editor” section, and submit your "Accept" recommendation.

Reviewer #1: (No Response)

Reviewer #2: All comments have been addressed

2. Does this manuscript meet PLOS Global Public Health’s publication criteria? Is the manuscript technically sound, and do the data support the conclusions? The manuscript must describe methodologically and ethically rigorous research with conclusions that are appropriately drawn based on the data presented.

Reviewer #1: Yes

Reviewer #2: Yes

3. Has the statistical analysis been performed appropriately and rigorously?

Reviewer #1: Yes

Reviewer #2: Yes

4. Have the authors made all data underlying the findings in their manuscript fully available (please refer to the Data Availability Statement at the start of the manuscript PDF file)?

Reviewer #1: Yes

Reviewer #2: Yes

5. Is the manuscript presented in an intelligible fashion and written in standard English?

Reviewer #1: Yes

Reviewer #2: Yes

6. Review Comments to the Author

Reviewer #1: Please see attached comments

Reviewer #2: Although there are always obstacles in conducting such research especially over such a long period of time, I congratulate the authors for doing this important research. I recommend accepting the paper.

7. PLOS authors have the option to publish the peer review history of their article (what does this mean?). If published, this will include your full peer review and any attached files.

**Do you want your identity to be public for this peer review?** For information about this choice, including consent withdrawal, please see our Privacy Policy.

Reviewer #1: No

Reviewer #2: **Yes: **Salaheddine Bendak

---

## [Editor Report · Decision Letter 2]

24 Jul 2024

Predicting Blood Lead in Uruguayan Children: Individual- vs Neighborhood-Level Ensemble Learners

PGPH-D-23-01421R2

Dear Dr. Frndak,

We are pleased to inform you that your manuscript 'Predicting Blood Lead in Uruguayan Children: Individual- vs Neighborhood-Level Ensemble Learners' has been provisionally accepted for publication in PLOS Global Public Health.

Best regards,

Meghnath Dhimal, Ph.D.

Academic Editor